# High Throughput Sequencing for Clinical Tuberculosis: An Overview

**DOI:** 10.3390/pathogens11111343

**Published:** 2022-11-14

**Authors:** Tara E. Ness, Andrew DiNardo, Maha R. Farhat

**Affiliations:** 1Division of Pediatric Infectious Diseases, Baylor College of Medicine/Texas Children’s Hospital, Houston, TX 77030, USA; 2Global TB Program, Baylor College of Medicine/Texas Childrens Hospital, Houston, TX 77030, USA; 3Harvard Medical School Biomedical Informatics and Pulmonary and Critical Care Massachusetts General Hospital, Boston, MA 02115, USA

**Keywords:** tuberculosis, *Mycobacterium tuberculosis*, sequencing, high throughput sequencing, next generation sequencing

## Abstract

High throughput sequencing (HTS) can identify the presence of *Mycobacterium tuberculosis* DNA in a clinical sample while also providing information on drug susceptibility. Multiple studies have provided a context for exploring the clinical application of HTS for TB diagnosis. The workflow challenges, strengths and limitations of the various sequencing platforms, and tools used for analysis are presented to provide a framework for further innovations in the field.

## 1. Introduction

High throughput sequencing (HTS), also known as next generation sequencing (NGS) can identify the presence of *Mycobacterium tuberculosis* DNA in a clinical sample while also providing information on drug susceptibility, making it an exciting prospect for the future of tuberculosis (TB) diagnostics. The World Health Organization’s END TB Strategy, originally passed in 2014, aims to end the TB pandemic by 2035 via three main indicators: reducing TB deaths by 95%, decreasing new cases by 90%, and ensuring families do not face catastrophic financial costs due to tuberculosis [1]. HTS has the potential to make advancements on all three of these indicators by providing (1) cost-efficient evaluation for TB from a range of patient clinical samples, (2) expedited information on drug susceptibility for those diagnosed with TB, and (3) distinguishing between infection relapse vs. reinfection [2]. The clinical application of these technologies for tuberculosis is an exciting prospect that may overcome many diagnostic and treatment challenges currently faced.

Worldwide, several approaches to TB diagnostics are used with phenotypic drug susceptibility testing (pDST) from sputum samples as the gold standard. Xpert MTB/RIF and Xpert Ultra (Cepheid, Sunnyvale, CA, USA) are now routinely used on patient clinical samples to provide molecular based rapid diagnostic testing for both *Mycobacterium tuberculosis* complex and rifampin resistance. In those with advanced HIV disease, urine TB-LAM has also been recommended. HTS, in contrast to these other diagnostic approaches, offers the ability to identify multiple potential drug mutations from a clinical sample in 1–2 days depending on the facility’s set up. By expanding our ability to identify multiple drug mutations at the point of diagnosis, we can avoid initiating a patient on ineffective treatment leading to unnecessary delays in proper care. Currently, HTS is being implemented as an adjunct to already employed diagnostics but has the capacity to become the first-line diagnostic approach, especially in certain populations. 

HTS refers to the ability to sequence multiple DNA fragments in parallel, allowing information to be obtained in a more rapid and cost-effective way compared to the original Sanger sequencing. DNA can be sequenced in short fragments, long fragments, or the entirety of the genome based on the instrument used, the clinical use and the sample characteristics (DNA quality, quantity, etc.). Whole genome sequencing (WGS) refers to the entirety of the genome being sequenced and is currently generally performed after culture to provide sufficient *M. tuberculosis* DNA for sequencing to occur. Targeted HTS (tHTS) first amplifies specific parts of a genome that are of interest and can be done on culture isolates or directly from clinical specimens (sputum, gastric aspirate, etc.) requiring less initial *M. tuberculosis* DNA input since amplification increases the quantity of DNA prior to it being loaded onto a sequencing platform. While these technologies represent new opportunities in the field of TB diagnostics, care must be taken to not only select the correct platform for the intended use but also ensure that proper infrastructure is in place to use the technology, especially in high TB prevalence and lower-resource settings where the need is greatest. Understanding the limitations of the variety of technologies available and the laboratory and bioinformatic infrastructure needs for clinical application is paramount to ensuring appropriate use and cost-efficiency when applying this technology for TB diagnosis.

## 2. Whole vs. Targeted Sequencing for TB

WGS can provide a wealth of information, including for TB outbreak investigation [3] and drug resistance [4], and has been used in several studies seeking to define genomic epidemiology of TB [5,6]. While previously performed only on TB culture isolates (representing a fraction of the pathogen population in host), WGS has now been applied directly to clinical specimens [7] and can provide more information on genetic diversity than a culture isolate [8]. A systematic analysis of over 40,000 isolates from 45 countries utilizing whole genome sequencing data against pDST showed a pooled sensitivity of more than 80% for rifampicin, isoniazid, ethambutol, fluoroquinolones, and streptomycin and specificity over 95% for all drugs except ethionamide, moxifloxacin and ethambutol [9]. Another study showed WGS of *M. tuberculosis* isolates to be 93% in agreement with pDST when performed on multi-drug resistant (MDR) or extremely drug resistant (XDR) TB patients [10]. Barriers to the wide scale implementation of WGS in clinical setting includes the high cost of WGS (141 to 277 USD per sequence) which has been decreasing [11] and the need for significant infrastructure for bioinformatic analysis [12]. The use of tHTS sequencing for *M tuberculosis* has been growing steadily and has advantages and disadvantages over WGS. Although it does not provide information on the entirety of the genome, careful primer design to investigate specific areas leads to greater depth of coverage for less resources. Several research studies have designed “in-house” multiplex polymerase chain reaction (PCR) panels for use on culture isolates [13,14] and clinical specimens such as sputum or bronchial aspirates [15,16,17]. The Deeplex^®^ Myc-TB assay (Genoscreen, Lille, France) is the most well-described commercially available targeted sequencing assay in the peer-reviewed literature. Deeplex is usable directly on sputum samples [18,19,20] but has also recently been applied successfully to stool samples [21]. Table 1 provides an overview of selected studies utilizing tHTS for tuberculosis, including the sequencing platform used, clinical specimen, and location where study was performed. Due to the cost and infrastructure needed for WGS, targeted approaches are currently more easily applicable for clinical settings as they do not require specimen culture and are often less bioinformatics intensive. 

Most studies have been conducted in Africa (Ghana, Kenya, Uganda, Zambia, Djibouti, Eritrea, Botswana, South Africa, Congo), with other studies being fairly heterogenous in their geographic coverage including Europe (Italy, Germany, Moldova, Spain) and Asia (China, Hong Kong, India). One study was performed in the United States [14] and no studies were in Latin or South America. In addition to the Deeplex Myc-TB commercial assay, several studies have evaluated the Next Gen-RDST/SMOR assay developed by Colman et al., 2016 [22], which utilizes amplicon targets and automated data analysis scripts to predict drug resistance.

## 3. Workflow

HTS workflows for TB are generally similar in their approach (Figure 1). Patient samples are collected and to date have included sputum, stool, bronchial aspirate, gastric aspirate, lymph node tissue, and bone marrow tissue for successful downstream sequencing (Table 1). Early morning samples of sputum, gastric aspirates, or bronchial aspirates are preferred as they have been shown to have the highest yield [31]. When intended for culture, multiple samples are ideal to increase the chance of enough viable organism for growth. If not used immediately, samples are stored at −80 °C. Sputum, specifically, is usually stored with cetylpyridinium chloride (CPC), ethanol, or commercially available products such as intended to help preserve *M. tuberculosis* DNA, decrease contamination, and improve culture yield [32,33]. DNA extraction on sputum or culture isolates has used a variety of methods and kits including ethanol precipitation with pre-treatment steps [7,27,34], the Maxwell 16 FFPE Tissue LEV DNA Purification Kit (Promega, Madison, WI, USA) [18,19,20], QIAamp DNA-Mini-Kit (Qiagen) [35], and the Roche Cobas Amplicor extraction kit [13]. One study extracted *M. tuberculosis* DNA from stool for sequencing and used the MPFast DNA Extraction Kit for Soil [21] while another study extracted from FFPE tissues and utilized the FFPE DNA kit (Taipu Biosciences Co., Ltd., Beijing, China) [24]. Library preparation can take varying amounts of time depending on the kit used and goal of sequencing. Some kits or steps can be used interchangeably between platforms and automation of library prep continues to decrease the time and cost burden [36]. After DNA extraction and throughout library preparation steps, the quality and quantity of *M. tuberculosis* DNA can and should be assessed using a variety of tools. When available, an automated electrophoresis system can provide information about DNA fragment sizes as well as quantity. When electrophoresis is not available, the combination of fluorometric quantification and gel electrophoresis can provide information on DNA fragment size, quality and quantity at a more affordable cost (with protocols per specific manufacturers). Spectrophotometry can provide information about sample purity which is especially helpful when troubleshooting downstream issues encountered with sequencing. Library preparation is usually platform-specific though there have been studies using library preparation kits from one manufacturer successfully on another platform with minimal protocol adjustments [18]. Cell lysis and homogenization of the sample is imperative to be able to access *M. tuberculosis* DNA that may be heterogeneously distributed in the clinical sample, however excessive sample disruption may also lead to DNA fragmentation. A variety of lysis buffers are available and mechanical disruption in the form of glass or zirconia beads is often utilized. Purification of samples from reagent contaminants that may interfere with successful sequencing often uses paramagnetic beads at multiple steps during the preparation process. Care must be taken as this can also decrease overall *M. tuberculosis* DNA yield as some DNA is lost in the purification steps from being “washed away” with the intended contaminants, making fluorometric quantification at different steps essential to ensure ample DNA is present for subsequent steps.

Multiplexing is a cost-effective means to run multiple samples at one time by giving individual samples a “barcode” that can be identified during the bioinformatics analysis. While this approach increases the efficiency by which large numbers of samples can be processed, it also decreases the amount of data that can be produced for each individual sample so must be taken into consideration given the research or clinical question or the data throughput of the pipeline.

## 4. Sequencing Platforms

The most well-known and heavily utilized platform for short-read sequencing is Illumina’s line of technologies (e.g., iSeq, MiSeq, and NextSeq) which uses a “sequencing by synthesis” approach where terminator nucleotides are incorporated into growing DNA strands and can be detected and then reassembled to a reference sequence. Ion Torrent (ThermoFisher Scientific) uses a similar “sequencing by synthesis” approach but detects pH changes released during DNA synthesis to determine the nucleotide sequence. Long-read sequencing approaches include Pacific Biosciences single-molecule real time (SMRT) sequencing approach where DNA fragments are ligated into loops by specific adaptor sequences and immobilized in channels where fluorescent light is detected as phospholinked nucleotides are incorporated. Lastly, Oxford Nanopore’s newer nanopore technology measures ionic current fluctuations as single-stranded nucleic acids pass through biological nanopores. Table 2 provides a summary of the benefits and limitations with each approach alongside selected publications that utilized each platform for *M. tuberculosis* DNA sequencing.

## 5. Sequencing Data Analysis and TB Resistance Platforms

Sequencing platforms generate data in the form of FAST or FASTQ files, with FASTQ files including information on the quality of the nucleotide that was determined reported as a Phred score [45]. At present, there is one commercially available automated web app for tHTS of *M. tuberculosis* (Deeplex Myc-TB) however others have been reported to be in development [46]. Without an automated setup, there are a myriad of ways to manually set up a bioinformatics pipeline to analyze results. FAST or FASTQ files of *M. tuberculosis* nucleotide sequences are used with a variety of software programs (Minimap2, BWA) to align the sequence to a reference creating a SAM or BAM file. BAM files are smaller in size requiring less data storage space but are a binary file format and therefore unable to be interpreted by the user in the raw text format. Basic quality control and sequence trimming (using programs such as FastQC and Trimmomatic) occur prior to sequence alignment to a reference (usually H37Rv, NC_000962.3), with some approaches also aligning the generated sequences to human and bacterial genomes to classify and filter out non- *M. tuberculosis* sequences more systematically. More advanced quality control, including sorting, duplicate control, and indexing can be done prior to variant calling, with the most popular software programs used for variant calling of *M. tuberculosis* being Pilon [47], vSNP [48], SNiPgenie [49], MTBSeq [50], and BovTB [51]. Variant filtering and annotation occurs at this step as well. Many public health laboratories have developed their own software programs or developed a customized workflow utilizing aspects of existing tools. While the most popular database for determining drug resistance for tuberculosis is the World Health Organization’s resistance database [9], others are available including KvarQ [52], PhyResE [53], MTBSeq [50], and Mykrobe [54]. Laboratories often have a dedicated bioinformatics specialist to develop the pipeline based on the specific needs of the research group especially regarding data storage. The benefits of an in-house bioinformatics pipeline are the ability to customize your setup for the research questions you are interested in and provide regular updates as technology progresses. Automated pipelines, while convenient, require waiting for the developers to release updates and may not always keep up as quickly with the changing research landscape. Due to the heterogeneity in data analysis approaches, comparison of studies can be challenging.

## 6. Future Directions

The use of genotypic DST for clinical decision making is still on the horizon but represents an exciting area of future research. Currently, the main means to determine genotypic drug susceptibility using HTS is via culture growth (WGS or tHTS) or directly from sputum (tHTS), with culture growth delaying clinical utility for ~2–4 weeks. Using tHTS directly on clinical samples would provide faster diagnostic decisions but can also be limiting as it only selects for certain regions to be amplified. Because of this, assays would need to be consistently updated to target new regions that are discovered to be associated with drug resistance in order to stay relevant. This would also need to be incorporated into bioinformatics pipelines to interpret the sequencing data from these new targets. The ideal assay would provide information on all anti-tuberculous drugs currently used in clinical practice, have a coverage depth allowing for identification of mixed infections, and have an automated, user-friendly bioinformatics pipeline with regular software updates incorporating new drug mutations that are discovered. While lineage information and strain typing may not yet be used for clinical treatment decisions, it helps with surveillance and may help us explore poorly understood concepts of why certain cases require different treatment lengths or are more likely to have relapse after receiving adequate treatment. Combining information about tuberculosis strains and immune phenotypes in the host may shed light on the enigma of tuberculosis disease progression.

At present, studies have investigated the sensitivity and specificity of DST determined from HTS compared to culture-based DST, with limited exploration of clinical outcomes had HTS alone been used for planning drug regimens. As gene causes of resistance are explored further, the accuracy of prediction will improve and HTS is likely to increase in importance as a tool to guide drug regimen composition in TB care. As costs decrease and portability improves, NGS will be able to be implemented in settings where the tuberculosis burden is high and resources are more limited. The use of HTS directly on clinical samples, including the exploration of newer sample types such as stool, is a promising direction to optimize the use of sequencing for diagnosing tuberculosis in its diverse clinical presentations.

## 7. Conclusions

Overall, HTS in the clinical setting is an exciting and fast evolving area. It offers the potential for rapid diagnosis and information on drug resistance of *M. tuberculosis* earlier than pDST options and with expanded information than current molecular-based methods such as Xpert and Xpert Ultra. As bioinformatics becomes more automated and accessible, HTS is expected to expand to the clinical setting and augment, if not supplant, current methods.

## Figures and Tables

**Figure 1 pathogens-11-01343-f001:**
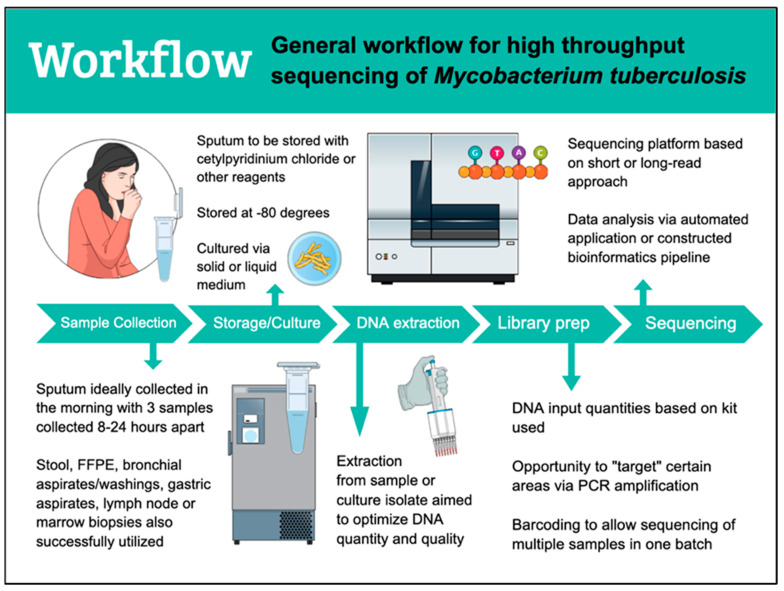
Workflow for high throughput sequencing of tuberculosis.

**Table 1 pathogens-11-01343-t001:** Overview of tHTS studies directly on clinical specimens for *Mycobacterium tuberculosis* DNA.

Author/Year	Location	tHTS Kit	Index Test Sample	Platform	Reference Standard/Comparisons
**Cabbibe 2020 [18]**	Italy	Deeplex Myc-TB	Sputum	MinION	Deeplex Myc-TB amplicons on MiniSeq
**Kambli 2021 [23]**	Mumbai, India	Deeplex Myc-TB	Sputum, isolates	iSeq	pDST from MGIT; LPA; pyrosequencing
**Kayomo 2020 [19]**	Congo	Deeplex Myc-TB	Sputum	MiSeq	Xpert MTB/RIF
**Colman 2019 [14]**	United States	SMOR assay [24]	Isolates	MiSeq, iSeq	WGS (MiSeq and iSeq)
**Tafess 2020 [13]**	Hong Kong, Ethiopia	In-house PCR (19 gene targets)	Isolates	MiSeq, MinION	pDST from MGIT
**Gliddon 2021 [25]**	Kwazulu, South Africa	In-house RPA (isothermal)	Isolates	MinION	WGS (HiSeq), pDST from MGIT and solid agar
**Sibandze 2022 [21]**	Eswatini, Germany	Deeplex Myc-TB	Stool	NextSeq	pDST from MGIT
**Wang 2019 [26]**	Botswana	SMOR assay [24]	Isolates	MiSeq	pDST on MGIT; LPA, Xpert MTB/RIF
**Mariner-Llicer 2021 [27]**	Spain	In-house assay (11 gene targets)	Sputum, isolates	MinION	WGS (MiSeq)
**Mesfin 2021 [20]**	Eritrea	Deeplex Myc-TB	Sputum	MiniSeq	Xpert MTB/RIF
**Tagliani 2017 [28]**	Djibouti	Deeplex Myc-TB	Sputum	MiniSeq	WGS (HiSeq) from isolates
**Song 2022 [24]**	China	In house PCR (11 gene targets)	FFPE tissues	Ion Proton	pDST from microtitre plate
**Chan 2020 [15]**	Hong Kong	In-house PCR (10 gene targets)	Bronchial aspirate, LN, sputum, bone marrow	MinION, MiSeq	pDST from MGIT; LPA
**Rowneki 2020 [16]**	Ghana, Kenya, Uganda, Zambia	In house PCR (17 gene targets)	Sputum	MiSeq	Sanger sequencing on subset of samples
**Colman 2016 [22]**	Moldova	SMOR assay	Sputum	MiSeq	pDST from MGIT
**Colman 2015 [29]**	Moldova	SMOR assay	Sputum	MiSeq	pDST from MGIT
**Zhao 2022 [17]**	Shanghai, China	In-house PCR (7 gene targets)	Sputum	GridION	Sanger sequencing, pDST
**Jouet 2021 [30]**	Djibouti, Congo	Deeplex Myc-TB	Sputum	MiSeq	WGS, pDST from Löwenstein–Jensen or Middlebrook 7H11 agar

**Table 2 pathogens-11-01343-t002:** Sequencing platforms used for HTS of *M tuberculosis* [37] including selected studies using particular platforms.

Platforms	Characteristics	Pros	Cons	Studies
**Illumina** **iSeq** **MiniSeq** **Miseq** **Nextseq** **HiSeq** **NovaSeq**	Short-read (2 × 150 bp, MISEQ 2 × 300 bp) Run time 4–72 h	Low error rate (99.9% accuracy) Platforms vary in low to high throughput	Difficulty in sequencing repetitive regions [38] Per base error rate increases with read length (trimming can improve) Long run times	Sibandze 2021 [21]; Kambli 2021 [23]; Kayomo 2020 [19]; Wang 2019 [26]; Colman 2019 [14]; Mesfin 2021 [20]; Tagliani 2017 [28]; Vogel 2021 [11]
**Thermo fisher Ion torrent** **Proton** **PGM** **S5**	Short-read (200–400 bp) Run time 3–24 h	Short run time Low error rate	Low performance on homopolymer regions High cost per sample	Daum 2012 [39]; Pavel 2016 [40]
**Pacbio** **RSII** **Sequel**	Long-read (10–60 kb) Run time up to 20 h	Short run time, long read length	High cost, high error rate (single base pair deletions most common, can improve with increased depth)	Lee 2019 [41]; Ley 2019 [42]
**Oxford Nanopore** **Minion** **Gridion** **Promethion**	Long-read (900 kb +) Run time 30 min to 48 h	Short run time, long read length Increasing portability	Historically higher error rate (>98% reported accuracy with newer technology) [43,44]	Gliddon 2021 [25]; Mariner-Llicer 2021 [27]; Zhao 2022 [17]

## Data Availability

Not applicable.

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
