# Peer review of "High Throughput Sequencing for Clinical Tuberculosis: An Overview"

_pathogens, 2022, doi:10.3390/pathogens11111343_

Round 1

Reviewer 1 Report

The manuscript of Ness, DiNardo, and Farhat entitled “High Throughput Sequencing for Clinical Tuberculosis: An Overview” discusses some aspects of the various sequencing platforms and tools involved in clinical tuberculosis. This interesting manuscript may contribute to the study, identification, and treatment of tuberculosis, especially when an overview of the subject is needed. Below I describe some considerations that could be made by the authors to improve the manuscript.

1. The authors say (lines 82-85) that studies have been conducted in Africa, Europe, Asia, and the United States. It would be interesting for the authors to comment if there are similar studies in Latin America. If so, it should be discussed. If not, it should be pointed out.

 2. In line 107, I think it would be more appropriate to use “- 80 °C” instead of “-80 degrees.” Similar observation for Figure 1.

3. In lines 120-124, the authors described the use of fluorometric quantification and spectrophotometry when electrophoresis is not available. Could the authors include here references that present protocols for these two techniques?

 4. In lines 133-135, the authors mentioned that “Care must be taken as this can also decrease overall M. tuberculosis DNA yield…”. What care needs to be taken and why? Could the authors discuss this a little more?

5. It is not clear to me if the topic “6. Use of HTS for Clinical TB Diagnosis” is a conclusion or not. Could the authors include a separate conclusion briefly resuming what was discussed during the review?

6. I suggest that authors consider including the following references.

1.        Gamaarachchige, G.; Suminda, D.; Bhandari, S.; Won, Y.; Goutam, U.; Kanth, K.; Son, Y.; Ghosh, M. High-Throughput Sequencing Technologies in the Detection of Livestock Pathogens , Diagnosis , and Zoonotic Surveillance. Comput. Struct. Biotechnol. J. 2022, 20, 5378–5392, doi:10.1016/j.csbj.2022.09.028.

2.        Gray, A.N.; Koo, B.M.; Shiver, A.L.; Peters, J.M.; Osadnik, H.; Gross, C.A. High-Throughput Bacterial Functional Genomics in the Sequencing Era. Curr. Opin. Microbiol. 2015, 27, 86–95, doi:10.1016/j.mib.2015.07.012.

3.        Strobel, E.J.; Yu, A.M.; Lucks, J.B. High-Throughput Determination of RNA Structures. Nat. Rev. Genet. 2018, 19, 615–634, doi:10.1038/s41576-018-0034-x.

4.        Pallen, M.J.; Loman, N.J.; Penn, C.W. High-Throughput Sequencing and Clinical Microbiology: Progress, Opportunities and Challenges. Curr. Opin. Microbiol. 2010, 13, 625–631, doi:10.1016/j.mib.2010.08.003.

Author Response

  1. The authors say (lines 82-85) that studies have been conducted in Africa, Europe, Asia, and the United States. It would be interesting for the authors to comment if there are similar studies in Latin America. If so, it should be discussed. If not, it should be pointed out.

     We are happy to make this change.

  1. In line 107, I think it would be more appropriate to use “- 80 °C” instead of “-80 degrees.” Similar observation for Figure 1.

      We are happy to make this change.

  1. In lines 120-124, the authors described the use of fluorometric quantification and spectrophotometry when electrophoresis is not available. Could the authors include here references that present protocols for these two techniques?

      These are usually done based on the instructions by the specific manufacturer and there are multiple platforms/systems available for this. We can add this line in the manuscript if the reviewer would like.

  1. In lines 133-135, the authors mentioned that “Care must be taken as this can also decrease overall M. tuberculosis DNA yield…”. What care needs to be taken and why? Could the authors discuss this a little more?

      We are happy to make this change. We can add some information about how DNA yield can be lost in these steps since DNA is meant to bind to beads while contaminants are washed off, however some DNA can also be washed off if it does not bind 100% effectively to the beads.

  1. It is not clear to me if the topic “6. Use of HTS for Clinical TB Diagnosis” is a conclusion or not. Could the authors include a separate conclusion briefly resuming what was discussed during the review?

      Yes, we can add an additional paragraph at the end summarizing the review.

  1. I suggest that authors consider including the following references.
  2.        Gamaarachchige, G.; Suminda, D.; Bhandari, S.; Won, Y.; Goutam, U.; Kanth, K.; Son, Y.; Ghosh, M. High-Throughput Sequencing Technologies in the Detection of Livestock Pathogens , Diagnosis , and Zoonotic Surveillance. Comput. Struct. Biotechnol. J. 2022, 20, 5378–5392, doi:10.1016/j.csbj.2022.09.028.
  3.        Gray, A.N.; Koo, B.M.; Shiver, A.L.; Peters, J.M.; Osadnik, H.; Gross, C.A. High-Throughput Bacterial Functional Genomics in the Sequencing Era. Curr. Opin. Microbiol. 2015, 27, 86–95, doi:10.1016/j.mib.2015.07.012.
  4.        Strobel, E.J.; Yu, A.M.; Lucks, J.B. High-Throughput Determination of RNA Structures. Nat. Rev. Genet. 2018, 19, 615–634, doi:10.1038/s41576-018-0034-x.
  5.        Pallen, M.J.; Loman, N.J.; Penn, C.W. High-Throughput Sequencing and Clinical Microbiology: Progress, Opportunities and Challenges. Curr. Opin. Microbiol. 2010, 13, 625–631, doi:10.1016/j.mib.2010.08.003.

      We are willing to integrate these references into the paper but would appreciate the reviewer indicating specific areas where these references would be helpful. We tried to have the review focus on the use of sequencing specifically for M. tuberculosis (with references being papers focused on this pathogen) and the above references provided by the reviewer appear to be related to non-M. tuberculosis pathogens.

Reviewer 2 Report

Although the introduction shows an acceptable scientific background, I think it addresses some important points poorly and lacks the theoretical bases with respect to the classical methods used to detect TB drug resistance such as the phenotypic standard technique and molecular methods, and their limitations. The review also lacks important data in the bioinformatic analysis, especially in the data pre-processing, quality control and annotation.

I have several important and consistent comments that should be addressed:

1.    The authors should include in the introduction section the traditionally employed molecular approaches and phenotypic susceptibility testing, and better describe their limitations in the context of NGS techniques.

2.    I would suggest the authors to have a clear explanation of the steps of bioinformatics pipeline of next generation sequencing of Mycobacterium tuberculosis including basic and advanced QC and trimming, variant calling and filtering, and annotation. A comprehensive presentation of the steps of a general bioinformatics pipeline should add value to this review (the most used packages).

3.    Please discuss about microorganism identification and filtering by alignment. What is the role of mapping with different reference genomes, including the human reference genome, viruses and bacterial genomes?4.    What are the advantages of the in house bioinformatic pipelines?

5.  Please change the Sanger sequencing platform with a next generation sequencing platform in the Figure 1.

6. Please discuss about the limited ability of nanopore sequencing platform from Oxford Nanopore Technologies (ONT) to detect variants in other unknown gene regions.

7.     Additionally, the future directions should be better represented in this review.

Author Response

Although the introduction shows an acceptable scientific background, I think it addresses some important points poorly and lacks the theoretical bases with respect to the classical methods used to detect TB drug resistance such as the phenotypic standard technique and molecular methods, and their limitations. The review also lacks important data in the bioinformatic analysis, especially in the data pre-processing, quality control and annotation.

 Thank you for this thoughtful and important feedback aimed at improving our manuscript in a constructive way. We hope we have addressed your concerns adequately.

I have several important and consistent comments that should be addressed:

  1. The authors should include in the introduction section the traditionally employed molecular approaches and phenotypic susceptibility testing, and better describe their limitations in the context of NGS techniques.

This is an excellent suggestion. We have added the following to the Introduction to better address this:

Worldwide, several approaches to TB diagnostics are used with phenotypic drug susceptibility testing (pDST) from sputum samples as the gold standard. Xpert MTB/RIF and Xpert Ultra (Cepheid, Sunnyvale, CA, USA) are now routinely used on patient clinical samples to provide molecular based rapid diagnostic testing for both Mycobacterium tuberculosis complex and rifampin resistance. In those with advanced HIV disease, urine TB-LAM has also been recommended. HTS, in contrast to these other diagnostic approaches, offers the ability to identify multiple potential drug mutations from a clinical sample in 1-2 days depending on the facility’s set up. By expanding our ability to identify multiple drug mutations at the point of diagnosis, we can avoid initiating a patient on ineffective treatment leading to unnecessary delays in proper care. Currently, HTS is being implemented as an adjunct to already employed diagnostics but has the capacity to become the first-line diagnostic approach, especially in certain populations.  

2.I would suggest the authors to have a clear explanation of the steps of bioinformatics pipeline of next generation sequencing of Mycobacterium tuberculosis including basic and advanced QC and trimming, variant calling and filtering, and annotation. A comprehensive presentation of the steps of a general bioinformatics pipeline should add value to this review (the most used packages).

Thank you for this feedback. Since clinical diagnosis of TB is mainly being applied with automated tools, we did not explore this area as detailed since it is more used in the research space. To address your concern, we have added “Basic quality control and sequence trimming (using programs such as FastQC and Trimmomatic) occur prior to sequence alignment to a reference (usually H37Rv, NC_000962.3), with some approaches also aligning the generated sequences to human and bacterial genomes to classify and filter out non- M. tuberculosis sequences more systematically. More advanced quality control, including sorting, duplicate control, and indexing can be done prior to variant calling , with the most popular software programs used for variant calling of M. tuberculosis being Pilon [47], vSNP [48], SNiPgenie [49], MTBSeq [50], and BovTB [51]. Variant filtering and annotation occurs at this step as well.”

We are open to creating a figure outlining these steps if the reviewer feels this would be helpful.

  1. Please discuss about microorganism identification and filtering by alignment. What is the role of mapping with different reference genomes, including the human reference genome, viruses and bacterial genomes?

Thank you, we have added “Some approaches also align generated sequences to human and bacterial genomes in order to classify and filter out non- M. tuberculosis sequences more systematically.”

  1. What are the advantages of the in house bioinformatic pipelines?

Thank you- we have added “The benefits of an in-house bioinformatics pipeline are the ability to customize your setup for the research questions you are interested in and provide regular updates as technology progresses. Automated pipelines, while convenient, require waiting for the developers to release updates and may not always keep up as quickly with the changing research landscape.”

  1.  Please change the Sanger sequencing platform with a next generation sequencing platform in the Figure 1.

Thank you for pointing this out. While we agree a more specific HTS platform would be ideal, we also did not want to endorse a specific platform via our Figure and also had limited art available for this purpose.

  1. Please discuss about the limited ability of nanopore sequencing platform from Oxford Nanopore Technologies (ONT) to detect variants in other unknown gene regions.

Nanopore sequencing, specifically from Oxford Nanopore Technologies, utilizes long-read sequencing so is able to sequence the entirety of the genome including unknown gene regions to detect potential variants. Please clarify your comment so we can adequately address your concern.

  1. Additionally, the future directions should be better represented in this review.

Thank you for pointing this out. We have expanded this section but please let us know if there are particular areas we should represent that are not already mentioned.

Reviewer 3 Report

This concise yet comprehensive review describes the current methods of NGS technology for TB diagnostics and research. All available methods are properly mentioned and explained. I have only one consideration regarding the runtime of different NGS platforms: the authors did not mention the time taken by library preparation. Maybe I do not have enough experience, but it seems likely that modern Illumina prep takes less time than Oxford Nanopore, so the total time is comparable.

Author Response

Thank you for this thoughtful point! There are a number of different library preparation kits that can be used on a given system based on the clinical question, with some being interchangeable, so we did not use parameter in comparing the time of different platforms. We can discuss this in the paper and also reference this recent review article specifically on library preparation in sequencing.

J.F. Hess, T.A. Kohl, M. Kotrová, K. Rönsch, T. Paprotka, V. Mohr, T. Hutzenlaub, M. Brüggemann, R. Zengerle, S. Niemann, N. Paust,
Library preparation for next generation sequencing: A review of automation strategies, Biotechnology Advances, Volume 41, 2020.

Round 2

Reviewer 2 Report

I think the authors addressed my comments. Regarding the comment #6, please read this article: 

1. Zhao K, Tu C, Chen W, Liang H, Zhang W, Wang Y, Jin Y, Hu J, Sun Y, Xu J, Yu Y. Rapid Identification of Drug-Resistant Tuberculosis Genes Using Direct PCR Amplification and Oxford Nanopore Technology Sequencing. Can J Infect Dis Med Microbiol. 2022 Mar 28;2022:7588033. doi: 10.1155/2022/7588033. PMID: 35386470; PMCID: PMC8979720.

Author Response

Thank you for this article! It was very interesting. It's questionable why there were certain areas the Nanopore sequencer missed additional gene variants, but a potential reason is due to primer design, not the sequencing platform itself.

We are happy your points were addressed adequately and want to thank you again for all of your constructive feedback!